# Screening for Hyperthermophilic Electrotrophs for the Microbial Electrosynthesis of Organic Compounds

**DOI:** 10.3390/microorganisms10112249

**Published:** 2022-11-14

**Authors:** Rabja Maria Popall, Alenica Heussner, Sven Kerzenmacher, Pierre-Pol Liebgott, Guillaume Pillot

**Affiliations:** 1Aix Marseille Université, Université de Toulon, IRD, CNRS, MIO UM 110, 13288 Marseille, France; 2Center for Environmental Research and Sustainable Technology (UFT), University of Bremen, 28359 Bremen, Germany

**Keywords:** microbial electrosynthesis, hyperthermophile, hydrothermal vent, organic acids synthesis

## Abstract

Microbial electrosynthesis has recently emerged as a promising technology for the sustainable production of organic acids, bioplastics, or biofuels from electricity and CO_2_. However, the diversity of catalysts and metabolic pathways is limited to mainly mesophilic acetogens or methanogens. Here, eleven hyperthermophilic strains related to Archaeoglobales, Thermococcales, Aquificales, and methanogens were screened for microbial electrosynthesis. The strains were previously isolated from deep-sea hydrothermal vents, where a naturally occurring, spontaneous electrical current can serve as a source of energy for microbial metabolism. After 6 days of incubation in an electrochemical system, all strains showed current consumption, biofilm formation, and small organic molecule production relative to the control. Six selected strains were then incubated over a longer period of time. In the course of one month, a variety of metabolic intermediates of biotechnological relevance such as succinic acid and glycerol accumulated. The production rates and the promotion of specific metabolic pathways seemed to be influenced by the experimental conditions, such as the concentration of CO_2_ in the gas phase and electron acceptor limitation. Further work is necessary to clearly identify these effects to potentially be able to tune the microbial electrosynthesis of compounds of interest.

## 1. Introduction

In recent years, the discovery of microbes that feed on electrical current has prompted the development of microbial electrosynthesis (MES) as a novel branch of sustainable biotechnology. Directly opposed to the concept of a microbial fuel cell, the general aim of MES is to produce economically valuable compounds from industrial waste, such as CO_2_, via electrotrophic metabolism. This is achieved by growing the biocatalyst on the negatively charged cathode of an electrochemical cell that is ideally powered from a green electricity source. Originally coined by Rabaey and Rozendal in 2010 [1], production rates as well as the range of products achieved with this technique have continuously improved over the past decade [2]. However, one of the limitations remains the low diversity of known electrotrophic biocatalysts that can be applied in MES, leading to the limited range of metabolic pathways and products.

The capability to utilize electrical current as an energy source in metabolism is not well-studied, thus many organisms employing this strategy may fly under the radar of biotechnologists seeking to improve MES. Electrotrophy in the strict sense refers to the ability of direct electron uptake from solid inorganics that are conductive, but also includes indirect electron uptake mechanisms using chemical mediators (including H_2_ at the molecular level).

As both electroautotrophy and electrofermentation remain obscure, it is helpful to focus the search for novel biocatalysts on natural environments that may favor these processes. The extreme geophysiochemical conditions at hydrothermal vents have inspired biotechnological progress for decades, as they promote specialized microbial adaptions such as the famous heat-tolerant Taq-Polymerase [3] that has revolutionized molecular biology. Arguably, the most extreme habitat in hydrothermal vents is the chimneys. These structures are formed by mineral precipitation from the hydrothermal fluid that the vent spouts from the subseafloor, most commonly as a result of volcanic activity. The chimney is colonized by a variety of hyperthermophilic archaea and bacteria metabolizing vent fluid and seawater constituents such as H_2_, CO_2_, CH_4_, H_2_S, SO_4_, NO_3_, and small organic molecules. It is widely accepted that the metabolic activity of these communities is based on chemolithotrophy. However, recent studies have inspired propositions on electrotrophy as a different metabolic strategy based on a naturally occurring electrical current at hydrothermal vent chimneys.

In 2010, it was shown that the sulfide-rich chimneys at volcanic hydrothermal vents are electrically conductive and allow efficient electron transport between environments with different redox potentials [4]. In fact, there is a large redox difference between hydrothermal fluid and the ambient seawater [5]. While the former is characterized by oxidation of sulfide anions to elemental sulfur, the latter features reduction of O_2_ to hydrogen peroxide catalyzed by the chimney wall. The resultant potential gradient of up to −700 mV generates a continuous electron flow from the inside to the outside of the chimney [4]. In situ measurements confirm that the outer chimney surface features a highly negative potential [5].

Two recent pilot studies show that the flow of electrons across the chimney wall indeed serves as a metabolic energy source for certain hyperthermophilic communities [6,7]. Hyperthermophiles are especially promising with regard to MES, because their extreme growth optima minimize the risk of contamination in industrial environments [8]. Furthermore, increased metabolic rates, electrolyte conductivity, and substrate solubility at high temperatures may improve the overall efficiency of MES [9].

To scout for new potential production organisms, in this study, we screened phylogenetic representatives of the hyperthermophilic groups that Pillot et al. (2021, 2020) found in their electrotrophic enrichments, and evaluated their potential for MES. We assessed which taxa are able to form electrotrophic biofilms on the cathode in pure culture, which biomolecules these biofilms produce, and which soluble electron acceptors each strain prefers in doing so. A polarized cathode serving as the sole electron donor replicated the hydrothermal chimney wall, while suitable electron acceptors were supplied in the medium and CO_2_ gas or acetate was provided as a carbon source. Subsequent to an initial screening of all eleven strains, six promising ones representing each phylogenetic group were studied over a longer period of time, increasing the surface area of the cathode to enhance biofilm formation and electrosynthesis production rates.

## 2. Materials and Methods

### 2.1. Media and Precultures

Eleven hyperthermophilic strains associated with anaerobic Archaeoglobales (*Archaeoglobus profundus* (APr), *Archaeoglobus fulgidus* (AF), *Ferroglobus placidus* (FP), *Geoglobus ahanghari* (GA)), Thermococcales (*Thermococcus litoralis* (TL), *Thermococcus onnurineus* (TO), *Thermococcus kodakarensis* (TK)), methanogens (*Methanococcus thermolithotrophicus* (MT), *Methanocaldococcus jannaschii* (MJ)), and microaerophilic Aquificales (*Aquifex pyrophilus* (APy), *Aquifex aeolicus* (AA)) were obtained from the Mediterranean Institute of Oceanography, France. All of these strains are autotrophic, with the exception of heterotrophic Thermococcales. Anaerobic and microaerophilic precultures were established in serum bottles using a modified version of artificial seawater (ASM) anaerobic medium. The composition of the medium (Carl Roth, Germany) was the following: NaCl (20 g L^−1^), MgCl_2_·6H_2_O (5 g L^−1^), MgSO_4_·7H_2_O (0.3 g L^−1^), NH_4_Cl (0.3 g L^−1^), NaHCO_3_ (1 g L^−1^), CaCl_2_·2H_2_O (0.3 g L^−1^), KCl (0.3 g L^−1^), KH_2_PO_4_ (0.3 g L^−1^), yeast extract (0.2 g L^−1^), and Wolfe’s mineral solution (10 mL L^−1^). A combination of potential electron donors and acceptors was added to each preculture according to their metabolism. Precultures were incubated at 150 rpm in an incubator (Incubator 3032, GFL, Germany) at 80 °C, except *Methanococcus thermolithotrophicus* incubated at 55 °C.

### 2.2. Microbial Electrochemical System

Both screening (*n* = 3 per strain) and long-term (*n* = 1 per strain) experiments were conducted at 75 °C or 55 °C (Table 1) in sterile H-cells equipped with an anionic exchange membrane (AMI, Membrane Internationals, Ringwood, NJ, USA), a saturated calomel reference electrode (K10, Sensortechnik Meinsberg, Waldheim, Germany), and an iridium-tantalum oxide coated titanium mesh (2 cm × 2 cm) (Umicore, Hanau, Germany) anode (see Appendix A). For screening experiments, a 2.25 cm^2^ exposed surface of a graphite plate (Müller & Rössner GmbH & Co KG, Niederkassel, Germany) was used as cathode. In long-term experiments, a rolled-up rectangle of graphite felt (15.5 cm × 8 cm × 0.7 cm, Sigratherm GDF, SGL CARBON GmbH, Wiesbaden, Germany) wrapped in 0.25 mm titanium wire served as cathode (see Appendix A). Prior to the experiments, both cathode types were sonicated in 70% isopropanol for 10 min to increase the hydrophilicity of the surface for improved biofilm attachment by removing adsorbed particles. Graphite plates were subsequently sonicated for 10 min in deionized (DI) water, while the graphite felt was gently rinsed with DI water to avoid disintegration.

The H-cells were filled with 250 mL of ASM media and supplied with the target electron acceptor(s) sulfate, thiosulfate, nitrate, O_2_, and Fe(III) oxide (amorphous ferrihydrite) or acetate (Carl Roth, Karlsruhe, Germany) as a carbon source in case of the heterotrophic Thermococcales (Appendix A). The medium was continuously sparged using flow meters (Analyt-MTC, Müllheim, Germany) at 100 mL/min with N_2_:CO_2_ gas (80:20) for anaerobic conditions and N_2_:CO_2_:O_2_ gas (77.5:20:2.5) for microaerophilic conditions. In long-term experiments, CO_2_ was increased to 100% on day 13 in all H-cells to counteract an increasing experimental pH on *Aquifex pyrophilus* H-cells and keep comparative conditions. The gases were supplied through a hydrator to balance evaporation at high temperatures. Electrodes were poised at a fixed potential of −600 mV vs. SHE using a potentiostat (IPS Elektroniklabor, PGU-MOD-500mA, Münster, Germany) and the EcmWin software (IPS Elektroniklabor, Münster, Germany). This potential was chosen as it performed best in previous experiments on another thermophilic strain [10]. Chronoamperometry was performed to monitor current consumption at the cathode as a function of time. An abiotic control was run for a duration of 5 days prior to long-term experiments. Subsequently, the medium was inoculated to a maximum of the 2% of preculture. The screening experiments were maintained for 6 days, while the long-term experiments were maintained for 33 days. The medium was sampled daily to monitor the production and consumption of chemical species by the cultures.

**Table 1 microorganisms-10-02249-t001:** Summary of the screening experiment on the 11 (hyper)thermophilic strains. The soluble electron acceptor or carbon source was provided at 10 mM. The total charge consumption (in coulombs), the organic production, measured by HPLC, the electron acceptor consumption, measured by IC, and the cell density on the electrode, measure by qPCR, are presented here. NA: non available, the succinic and lactic acids could not be measure when thiosulfate was used as electron acceptor due to overlapping retention time with the HPLC method. Blue-colored values represent production, while red-colored values represent consumption. pH ranges were obtained from literature [11].

Strain		Electron Acceptors/Carbon Source	pH Range	Coulombs	HPLC (mM)						IC (mM)			qPCR (Log_10_ rRNA Copies·mL^−1^)
					Succinic Acid	Lactic Acid	Formic Acid	Acetic Acid	Propionic Acid	Isobutyric Acid	Nitrate	Sulfate	Thiosulfate	
*Archaeoglobus fulgidus*	AF	SO_4_, S_2_O_3_/CO_2_	5.5–7.5	202 ± 41	NA	NA	0.06 ± 0.05	0.02 ± 0.03	0.04 ± 0.05	0.07 ± 0.10	-	−3.59 ± 1.56	−6.98 ± 1.01	7.59 ± 0.87
*Archaeoglobus profundus*	APr	SO_4_, S_2_O_3_/CO_2_	4.5–7.5	272 ± 56	NA	NA	0.08 ± 0.03	-	0.09 ± 0.13	0.04 ± 0.06	-	−3.43 ± 3.40	−5.04 ± 2.46	7.45 ± 0.40
*Ferroglobus placidus*	FP	NO_3_, S_2_O_3_, Fe(III)_Am_/CO_2_	6.0–8.5	148 ± 38	NA	NA	0.01 ± 0.01	0.03 ± 0.04	0.01 ± 0.02	0.28 ± 0.45	−4.70 ± 7.71	-	−5.98 ± 4.68	5.18 ± 0.34
*Geoglobus ahangari*	GA	NO_3_, Fe(III)_Am_/CO_2_	5.0–7.6	113 ± 13	0.05 ± 0.04	-	0.06 ± 0.04	0.09 ± 0.15	0.05 ± 0.02	-	−9.53 ± 0.09	-	-	8.15 ± 0.26
*Thermococcus onnurineus*	TO	-/CO_2,_ acetate	5.0–9.0	113 ± 25	0.06 ± 0.05	0.06 ± 0.11	4.43 ± 0.06	−4.62 ± 2.16	0.09 ± 0.08	0.40 ± 0.68	-	-	-	7.62 ± 0.90
*Thermococcus kodakarensis*	TK	-/CO_2,_ acetate	5.0–9.0	78 ± 31	0.05 ± 0.08	0.06 ± 0.04	4.07 ± 0.03	−6.41 ± 0.85	0.09 ± 0.06	0.82 ± 0.34	-	-	-	5.35 ± 1.14
*Thermococcus litoralis*	TL	-/CO_2,_ acetate	6.2–8.5	77 ± 21	0.04 ± 0.03	0.18 ± 0.31	4.10 ± 0.07	−6.05 ± 0.73	0.01 ± 0.02	0.26 ± 0.44	-	-	-	4.57 ± 1.03
*Methanocaldococcus jannaschii*	MJ	-/CO_2_	5.2–7	70 ± 29	0.10 ± 0.13	0.03 ± 0.05	0.08 ± 0.05	0.04 ± 0.06	0.01 ± 0.02	0.31 ± 0.33	-	-	-	4.08 ± 0.46
*Methanococcus thermolithotrophicus*	MT	-/CO_2_	6.0–8.0	105 ± 40	0.04 ± 0.07	0.04 ± 0.05	0.08 ± 0.07	0.03 ± 0.04	0.02 ± 0.03	0.12 ± 0.21	-	-	-	3.88 ± 1.03
*Aquifex pyrophilus*	APy	NO_3_, O_2_/CO_2_	5.4–7.5	172 ± 105	0.03 ± 0.05	-	0.04 ± 0.02	0.27 ± 0.42	0.06 ± 0.04	0.42 ± 0.69	−10.14 ± 0.65	-	-	8.54 ± 0.94
*Aquifex aeolicus*	AA	S_2_O_3_, O_2_/CO_2_	5.4–7.5	96 ± 21	NA	NA	0.01 ± 0.01	0.04 ± 0.03	0.05 ± 0.03	0.05 ± 0.08	-	-	−6.68 ± 3.18	6.53 ± 1.21
*Abiotic control*	-	SO_4_, S_2_O_3_, NO_3_, O_2_, Fe(III)_Am_/CO_2,_ acetate		35 ± 5	NA	NA	-	−1.50 ± 0.21	0.01 ± 0.01	0.03 ± 0.01	−0.75 ± 0.23	−0.38 ± 0.47	−0.91 ± 0.52	-

### 2.3. Fluorescence Microscopy

After the end of experiments, cathodes were harvested and fixed in 4% glutaraldehyde in 0.1 M phosphate-buffered saline until further use. They were then analyzed for potential biofilms via fluorescence microscopy. The fixed electrodes were stained with 2 µg·mL^−1^ DAPI (4′,6-diamidino-2-phenylindole) and Acridine orange (Carl Roth, Germany) to target DNA/RNA and incubated in the dark for 30 min. The stained biofilms on the electrode materials were visualized using a Zeiss Microscope Axioscope 5/7 (Solid-State Light source Colibri 3 (Type RGB-UV)) and Microscopy Camera Axiocam 702 mono) (Zeiss, Germany) at 250× magnification (Objective ApoChrom 25×) under oil immersion, and subsequently the z-stacks were automatically captured with the motorized stage on the Zen software (Zeiss, Germany, version 3.0).

### 2.4. Biofilm Quantification by Means of qPCR

After the experiment, the cathodes containing the biofilm were taken from the bioelectrochemical reactor and sonicated for 10 min in 10 mL DI water suspension in order to detach the biofilm from the electrode surface. Furthermore, the 16S rRNA gene was partially amplified by the qPCR method in an Eco 48 real-time PCR System (PCRmax, United Kingdom), using the qPCRBio SyGreen 2x-Mix (Nippon Genetics Europe, Düren, Germany) and qPCR using archaeal- (913F: AGG AAT TGG CGG GGG AGC A and 1100R: BTG GGT CTC GCT CGT TRC C) and bacterial-specific (GML5F: GCC TAC GGA GGC AGC AG and Univ516R: GTD TTA CCG CGG CKG CTG RCA) 16S rRNA primers to amplify the V2-V3 region. Potential cross-contamination of the experiments was excluded by quantification of each strain with qPCR primers specific to each phylogenetic group, as presented previously [6,7].

### 2.5. Analysis of Produced Organic Compounds

Media samples were analyzed via high-performance liquid chromatography (HPLC) to establish the production of organic compounds over the course of the experiment. HPLC measurements were performed in an HPLC system (Alliance, Waters, Germany) with 2414 refractive and 2489 UV index detectors (Waters, Germany) equipped with an Aminex HPX-87H column (300 × 7.8 mm) with an 8mM H_2_SO_4_ eluent at 0.6 mL/min and column at 35 °C. Furthermore, media samples were analyzed via ion chromatography (IC) to assess the consumption of soluble electron acceptors. IC measurements were conducted on an IC system (Metrohm, Germany) equipped with a Metrosep A Supp-5 150/4.0 separation column and Metrosep A Supp 4/5 Guard/4.0 precolumn (35 °C, carbonate eluent with 6.4 mM Na_2_CO_3_, 2 mM NaHCO_3_, and 17 Vol-% Aceton at 0.6 mL/min). Succinic and lactic acids were excluded for strains supplied with thiosulfate as electron acceptor due to overlapping peaks in HPLC analysis. NMR analyses were performed at the end of the long-term experiments to identify and confirm the peaks obtained in HPLC. The NMR analyses were performed as previously described in Pillot et al. (2021a, 2020).

For all compounds produced and consumed, the coulombic efficiency (CE) was calculated to evaluate the ratio of produced and consumed electrons according to
$$\mathrm{CE}\left(\%\right)=\frac{F\cdot n_{e}\cdot \Delta \left[P\right]\cdot V}{\int_{t_{0}}^{t}i\left(t\right)\cdot \mathrm{dt}}\cdot 100$$
with F being the Faraday constant, n_e_ being the number of electron moles present per mole of product created (mol), Δ[P] being the variation in the concentration of the product between t_0_ and t (mol L^−1^), and V being the volume of the catholyte (L). The integration of current over time was approximated with the current difference between consecutive time points. Furthermore, the maximum production rates were averaged over 4 to 10 days per strain and compound.

## 3. Results

### 3.1. Screening Experiments

After the inoculation of the cathodic chamber of the individual H-cell system with the 11 (hyper)thermophilic strains, an increase in cathodic current was observed on each triplicate (Appendix A), with values of consumed charge higher than the abiotic control (35 ± 5 C) and with a mix of all the electron acceptors in the same concentration (Table 1). The total electrical charge consumed by each strain after 6 days varied from 70 ± 29 C for *Methanocaldococcus jannaschii* (MJ) to 272 ± 56 C for *Archaeoglobus profundus* (APr) (Table 1). Archaeoglobales tend to show higher current consumption in our condition, with values from 113 ± 13 C to 272 ± 56 C and average current density after 2 days between 0.18 mA∙cm^−2^ for *Geoglobus ahanghari* (GA) and 0.7 mA∙cm^−2^ for *Archaeoglobus fulgidus* (AF), but with a slow decrease over the last 3 days for the latter. They are followed by the Aquificales, with current consumption between 96 ± 21 C and 172 ± 105 C, showing the highest variation between triplicates and strains of the same group (Table 1). The current density from *Aquifex pyrophilus* (APy) tends to quickly increase up to 0.8 mA∙cm^−2^, and afterwards slowly decreases to 0.1 mA∙cm^−2^ over the 6 days. However, *Aquifex aeolicus* (AA) exhibited a relatively stable current consumption of around 0.2 mA∙cm^−2^ over the 6 days. Methanogens and Thermococcales presented a large variation in current density during the first day from 0.02 to 0.7 mA∙cm^−2^, which quickly stabilized to a current consumption between 0.05 to 0.25 mA∙cm^−2^ over the last 5 days, accounting for the lowest charge consumption between 70 ± 29 to 113 ± 25 C.

During the 6-day cultivation period, the production of organic acids and the consumptions of soluble electron acceptors and carbon source were followed by HPLC and IC, as presented in Table 1. The soluble electron acceptors, initially at a concentration of 10 mM, were partially or totally consumed depending on the strains, while only a neglectable part was reduced in the abiotic control (<1 mM). The highest consumptions over the 6-day period (expressed as concentration consumed) were observed for the nitrate by GA (9.5 ± 0.1 mM) and APy (10.1 ± 0.6 mM). The other *Archaeoglobus* species tend to prefer the consumption of thiosulfate (from 5.0 ± 2.5 to 7.0 ± 1.0 mM) over the consumption of their alternative electron acceptor, the sulfate, with 3.4 ± 3.4 to 3.6 ± 1.6 mM, with no sulfate consumption in some replicates. However, FP seems to balance between nitrate and thiosulfate consumption, depending on the replicate, with average values around 4.7 to 6.0 mM and a high standard deviation of 7.7 and 4.7, respectively. This could indicate a higher metabolic plasticity of FP compared to the other strains. Acetate was supposedly used as a nonfermentable carbon source for Thermococcales, and consumption between 4.6 and 6.4 mM could be observed during the 6 days of experiment with the three strains. The iron reduction was evaluated for FP and GA using a ferrozine assay but did not exhibit more than 0.2 mM reduced iron after 6 days. Unfortunately, the consumption of gaseous substrate, such as O_2_ for Aquificales and CO_2_ as the carbon source for all strains, could not sensibly be measured during these experiments.

Alongside the consumption of electrical current and electron acceptor, the production of small amounts of organic acids was measured. The most notable production was observed with the Thermococcales, converting acetate and/or CO_2_ into formic acid (up to 4.4 mM concentration increase over the 6-day period) and small amounts of isobutyric acid (up to 0.8 mM). The rest of the strains produced a small amount of isobutyric acid (up to 0.4 mM for APy and 0.3 mM for FP) and traces (<0.1 mM) of succinic, lactic, formic, acetic, and propionic acids.

All strains displayed cells on the cathode, however, only APy, AF, and GA formed dense biofilms covering the entire electrode surface (Figure 1). These strains also yielded the highest number of cells as quantified by qPCR (8.5 ± 1.0 log_10_ copies∙cm^−2^ for APy, 7.6 ± 0.8 log_10_ copies∙cm^−2^ for AF, and 8.15 ± 0.3 log_10_ copies∙cm^−2^ for GA) (Table 1). TO, APr, and AA developed spots of dense biofilms with cell quantities of 7.6 ± 0.9 log_10_ copies∙cm^−2^, 7.5 ± 0.4 log_10_ copies∙cm^−2^, and 6.5 ± 1.2 log_10_ copies∙cm^−2^, respectively. FP and TK showed relatively less growth, with cell quantities of 5.2 ± 0.3 log_10_ copies∙cm^−2^ and 5.3 ± 1.1 log_10_ copies∙cm^−2^, respectively. TL, MT, and MJ featured cell quantities below 4.6 log_10_ copies∙cm^−2^ and infrequent spots of cellular aggregations (Figure 1). Interestingly, FP and MT featured filamentous structures in all triplicates, reaching lengths of more than 50 µm (Figure 1) that are usually not observed in liquid culture. Inocula were estimated to bring in the total volume of the H-cells, up to 5 log_10_ cells for Aquificales and *Archaeoglobus* species and up to 3 log_10_ cells for the methanogens, FP and GA. This shows that there is cellular growth over the course of the experiment.

We can conclude that all strains can grow on a cathode in our conditions, consuming current and electron acceptors and producing small quantities of organic acids. To further evaluate their potential in electrosynthesis, six strains (GA, FP, AF, TO, MT, and APy) showing the highest current consumption or biofilm density and belonging to each phylogenetic group were selected to follow their growth on a cathode with higher surface area using a graphite-felt electrode and for an extended period of time of 33 days.

### 3.2. Long-Term Experiments

#### 3.2.1. Current Consumption and pH

In general, the current in long-term experiments was approximately 10 times higher compared to screening experiments due to the increased cathodic surface area (Figure 2B). The abiotic control showed a steep increase in current consumption directly after the start of the experiment, stabilizing around 0.05 mA∙cm^−3^ of volume of the cylindrical graphite felt cathode (estimated at 85 cm^3^). The values of the stabilized current consumption of abiotic controls were subtracted from the current of inoculated experiments before calculating the total electrical charge consumed ΔC and are presented in Figure 2B. Overall, MT showed the slowest increase in current consumption, reaching a ΔC_final_ of 3355 C. The current consumption by FP plateaued when the concentrations of thiosulfate decreased and recovered instantly when the thiosulfate was replenished. When the electron acceptor was switched to poorly crystalline and amorphous Fe(III) oxide on day 10, current consumption recovered only after few days.

APy showed the strongest increase in current consumption when O_2_ was added as the electron acceptor, reaching the far highest ΔC_final_ = 32,681. In the long-term experiments, two successive phases were studied. After 13 days of growth, the CO_2_ concentration was increased to 100% of the gas phase injected into each H-cell. This CO_2_ increase was initiated to overcome the alkalinization of the media beyond their maximum pH tolerance (Table 1) by acting as a carbonate buffer. The successive addition of HCl 25% (not shown on the graphs) did not allow to regulate the pH without adding a too-high amount and consequently increasing the salinity by addition of Cl- over the tolerance limit of the strains. Additionally, the media in APy were completely renewed on day 16, as the pH was increasing again over the tolerance limit. Surprisingly, the change in pH and/or CO_2_ concentration had a noticeable impact on the activity of most of the strains. GA, AF, and TO temporarily accelerated current consumption from day 13 on when CO_2_ was increased to 100% and reached final values of total electrical charge consumed of 9436, 11,701, and 15,138 C, respectively. In FP, current consumption decreased when CO_2_ was increased to 100%, reaching ΔC_final_ = 13,015 C.

#### 3.2.2. Organic Compound Productions

Similar to the screening experiments, all strains produced considerable amounts of organic compounds during the 33 days of experiment, presented in Figure 3. The various compounds measured over time by HPLC, and confirmed by ^1^H NMR on the final sample, included isopropanol, ethanol, glycerol, and formic, acetic, glycolic, pyruvic, propionic, succinic, isobutyric, and butyric acids. As mentioned in the Materials and Methods section, succinic and lactic acids were excluded from the results for strains supplied with thiosulfate as the electron acceptor (AF and FP) because of overlapping peaks in HPLC analysis. A small amount of alanine was measured by ^1^H NMR at the end of the experiment, but evolution of the concentration over the experiment could not be inferred. The concentration of these compounds, measured over the period at 20% CO_2_, at 100% CO_2_, and the maximum concentration reached for each strain and each compound during the experiment, are presented in Figure 3. Indeed, the concentration of some produced compounds decreased at some point, either through consumption, evaporation, or thermal degradation (Appendix A). Evaporation and thermal degradation were evaluated in an abiotic control and appear to affect only some organic acids (Appendix A), mainly acetate, propionate, and derivates of butyrate and valerate. The concentrations of some compounds, especially formic, acetic, propionic, and butyric acids, varied greatly between a CO_2_ supply of 20% and 100%.

Overall, the total concentration of products was 4 to 10 times higher at the end of the 20 days at 100% CO_2_ than at the end of 13 days at 20% CO_2_. This difference in production is not related to longer cultivation period, as higher production per day was observed at 100% CO_2_ (Appendix A). APy showed the maximum organic production, reaching up to 12.4 mM of formic acid, 3.6 mM of acetate, 1.5 mM of glycerol, 1.6 mM of succinic acid, and 0.8 mM of lactic acid on day 16. A maximum of 3.7 mM of isopropanol and 5.4 mM of propionic acid were obtained by FP, 1.2 mM of isobutyric acid by MT, 0.35 mM of Glycolic acid by AF, 0.85 mM of ethanol by TO, and 2.2 mM of butyric acid by GA.

#### 3.2.3. Coulombic Efficiency

The coulombic efficiency (CE, Figure 4) for each condition and at the end of the experiment allows to clearly see the difference in metabolism affected by the CO_2_ supply concentration. As main changes, GA converts most of the electron received from the electrode (86% in total) into ethanol, acetate, succinate, and butyrate at 20% CO_2_, and switches to isopropanol, propionate, and butyrate at 100% CO_2_. FP, with a total CE of 80%, starts diverting electrons into propionate only at 100% CO_2_. AF, with a total CE of 65%, produces mainly isopropanol at 20% CO_2_, but switches to more diverse production at 100% CO_2_. The Thermococcales (TO) produce mainly butyric and isobutyric acid at low CO_2_ and more diverse production at high CO_2_ percentage, with a lower total CE of 24% at the end. As no acetate consumption could be observed on TO, formate was added at day 5, 13, and 16 to act as an alternative carbon source to observe its effect on organic production and current consumption. Formate was consumed rapidly at the first addition, and the consumption slowed after increase at 100% CO_2_. MT, with a more diverse production at 20% CO_2_, produces mainly isopropanol at 100% CO_2_ to a total CE of 14%. Methane production would be expected by this methanogen but has not been quantified during the experiment. Moreover, the focus of this study is to find alternative electrosynthesis production, as methane electrosynthesis was already reported by other methanogens. Finally, APy, which exhibits the highest organic production, only transfers up to 17% of the consumed electron into this organic production and switches between ethanol and glycerol with the percentage of CO_2_.

#### 3.2.4. Soluble Electron Acceptors

Regarding the soluble electron acceptors, overall large concentrations of nitrate and thiosulfate were consumed. Similar to screening experiments, mostly distinct preferences for a certain electron acceptor could be observed (Figure 2C). APy did not consume a significant amount of nitrate, as observed in the screening experiment, and started consuming current only after the addition of O_2_ supply. GA started reducing nitrate during the first weeks and was forced to switch to Fe(III) reduction, observed by the black color of the hematite, originally red, when nitrate was totally depleted to allow a slower reduction rate and a longer growth before addition of electron acceptor. The same change in electron acceptor to Fe(III) was performed with FP to overcome its high thiosulfate consumption and H_2_S production, which was corrosive to the equipment.

#### 3.2.5. Biofilm Observation and Quantification

All strains exhibited growth on the cathodic graphite fibers, however, biofilm densities varied between strains and, depending on the strain, also on the localization in the rolled-up cylindrical graphite felt. AF featured the highest quantification by qPCR (6.56 log_10_copies∙cm^−3^) and formed a dense biofilm on the outside of the cathode between the carbon fibers and rich in EPS-like matrix (Figure 5). On the contrary, APy featured a dense biofilm (5.36 log_10_copies∙cm^−3^) on the inside of the cathode only (Figure 5), also with a rich EPS-like matrix. TO covered the entire cathode in an equally dense biofilm (5.36 log_10_copies∙cm^−3^), but as a monolayer around the fibers (Figure 5). For GA, 5.02 log_10_copies∙cm^−3^ were obtained. However, less growth could be observed under the fluorescence microscope with cellular aggregations dispersed on the inside of the cathode (Figure 5). MT yielded low quantities by qPCR (2.1 log_10_copies∙cm^−3^), but exhibited locally dense biofilms of very interesting morphology, especially on the outside of the cathode: the coccoid cells aggregated in a network of filamentous structures reminiscent of a spider web (Figure 5). For FP, 1.02 log_10_copies∙cm^−3^ 16s rRNA copies were obtained. Barely any growth was observed on the outside of the cathode, but on the inside spots of monolayers of cells could be found on the carbon fibers (Figure 5). No significant growth was observed in the planktonic phase during the experiments. The cell density obtained on the cathode at the end of the experiment cannot be easily correlated with organic production level or global current consumption. Indeed, the biofilms might have endured some electron acceptor starvation or stress during the last days of experiment, as observed on the stagnation of organic concentration after day 23 for most of the strains. Moreover, the high total coulombic efficiency for some strains, such as for FP at 100% CO_2_, suggests that little energy was diverted to biomass production, leading to low biofilm density.

## 4. Discussion

### 4.1. Electrotrophy of (Hyper) Thermophilic Strains

To date, only a few thermophilic microorganisms have been used in electrosynthesis. *Moorella thermoautotrophica* and *M. thermoacetica* have been previously immobilized on a cathode poised at −0.3 or −0.4 V vs. SHE at up to 70 °C to perform electrosynthesis of formate and acetate from CO_2_ [12,13]. *Moorella* and *Thermoanaerobacter* species were also enriched in communities on cathodes at 70 °C from anaerobic sludge [14]. *Kyrpidia spormannii* have been used on cathodes at 60 °C to produce polyhydroxyalkanoates (PHA) from CO_2_ [10]. Thus, the present investigation of 11 potential electrotrophs in pure culture significantly expands the panel of known (hyper)thermophilic electrotrophs.

The consumption of current, formation of biofilms on the cathode, uptake of supplied electron acceptors, and production of organic compounds suggest that all strains are capable of electrotrophy to a certain extent (Table 1). TL did not produce considerable concentrations of organics but displayed current consumption and growth on the cathode. On the contrary, several strains did not establish high cell densities on the cathode but presented considerable biomolecule production with current consumption rates that were distinguishable from abiotic controls. Moreover, the absence of planktonic growth confirms the dependence of the only energy source initially available in our system, the cathode.

### 4.2. Electrosynthesis Pathways to Valuable Organic Compounds

The growth of strains we observed on the cathode is complemented by a production of diverse organic acids, including the conventional electrosynthesis products, such as ethanol, isopropanol, formic acid, acetic acid, propionic acid, butyric acid, and isobutyric acid [2]. However, our (hyper)thermophilic strains also exhibited the production of unconventional products for electrosynthesis, such as glycerol, succinic acid, glycolic acid, lactic acid, and alanine, that have never been reported before from only CO_2_ as the carbon source, but some such as glycerol, succinate, or lactate could be produced from intermediates such as carbohydrates or other organic acids [15,16]. While some products are common to all strains, differences in production ratio can be observed depending on the CO_2_ fixation pathway or phylogenetic group. Understanding their metabolic pathway is then of interest. It has been suggested that electron overfeeding from a polarized cathode and excess of CO_2_ promote a release of intermediates from carbon metabolism due to an “overload” of respective pathways [6,17]. While the taxonomic groups we tested cover a wide spectrum of metabolic type and carbon fixation pathways, they employ similar catabolic and anabolic mechanisms [18].

#### 4.2.1. Carbon Fixation Pathways and Intermediates

Both the facultatively autotrophic Archaeoglobales and the obligatorily autotrophic methanogens use the reductive acetyl-CoA pathway for lithoautotrophic carbon fixation (Figure 6) [18]. The methyl branch of this pathway yields formate as an intermediate. Indeed, formic acid was produced in significant amounts by Archaeoglobales and methanogens in both screening and long-term experiments. When the methyl branch merges with the carboxyl branch, acetyl-CoA is yielded. This compound is a key intermediate linking catabolism and anabolism in most known organisms. Acetyl-CoA can then be converted to acetate by the action of a phosphate acetyltransferase (PTA) and an acetate kinase (ACK) producing one mole of ATP. Thus, acetic acid is likely ubiquitous in electron overfeeding conditions, as observed in acetogens [19]. The comparably low concentrations of acetic acid that we obtained may reflect the wide range of other routes that acetyl-CoA can take in both catabolism and anabolism (Figure 6).

Contrary to the archaeal Archaeoglobales and methanogens, the autotrophic bacterial Aquificales use the reverse tricarboxylic acid (TCA) cycle for lithoautotrophic carbon fixation (Figure 6). This pathway is the inversion of the amphibolic TCA cycle and has never been evaluated in microbial electrosynthesis before. Similar to its amphibolic equivalent, the reductive TCA cycle produces succinate as an intermediate. This may explain the high production of succinic acid in APy in the long-term experiment. Subsequent to initial CO_2_ uptake, this cycle results in acetyl-CoA formation and in the same catabolic and anabolic reactions as the reductive acetyl-CoA pathway [18].

#### 4.2.2. Acetyl-CoA and Further down the Road of Anabolism

As indicated above, acetyl-CoA can take a multitude of routes in the further metabolism. First, it may be interconverted to pyruvate by pyruvate:ferredoxin oxidoreductase (PFOR) [20,21]. Putative metabolic pathways of pyruvate in microbial electrosynthesis include isobutyrate production through the Ehrlich pathway [19], as well as glycerol production, such as in *Saccharomyces cerevisiae* or *Bacillus subtilis* by reduction of the glycolytic intermediate dihydroxyacetone phosphate to glycerol 3-phosphate (G3P) followed by dephosphorylation of G3P to glycerol [22,23]. Significant small amounts of dihydroxyacetone were detected by ^1^H NMR spectroscopy at the end of the experiments, probably as intermediates of this production. This compound has been shown to function as a compatible solute (osmolyte) in osmoregulation [24], which corresponds to the strongly increased pH during long-term experiments. The cathodic reaction continuously consumes H^+^, elevating pH and potentially creating osmotic stress in the biofilm in the long term. In APy, this was likely exacerbated by the production of OH^-^ by O_2_ reduction, resulting in the highest alkalinity and in the highest initial glycerol production out of all strains. Glycerol was produced and consumed by all strains during long-term experiments. A recent study showed that lactic acid and glycolic acid can be derived from glycerol in electrosynthesis [25], probably explaining the small amount of these compounds concomitant with glycerol production and consumption with most strains.

Next to these catabolic routes, acetyl-CoA is fed into the TCA cycle at the interface of catabolism and anabolism. It has been shown that the TCA cycle can be driven backwards by high partial pressures of CO_2_ [17], which may explain the overall higher concentrations of products such as formic acid, acetic acid, and isobutyric acid at a 100% supply of CO_2_ during long-term experiments (Figure 3 and Figure 4). An intermediate of the TCA cycle is succinate. Interestingly, succinic acid was produced mainly at a CO_2_ concentration of 20% in long-term experiments (Figure 3 and Figure 4). This suggests that the accumulation of this compound was impeded by a backward-driven TCA cycle, or that a more pH-neutral environment favors the conversion of succinic acid into other compounds. In biosynthesis, propionate may be derived from succinate [26]. In fact, propionic acid was mainly produced at 100% CO_2_ in long-term experiments. As succinate is an intermediate in all types of carbon metabolism presented, this would furthermore explain the wide diversity of strains producing propionic acid during long-term experiments. Likewise, isobutyric acid might have been produced from the reduction of succinate to butyrate and subsequent isomerization of butyrate to isobutyrate under anaerobic conditions [27]. Interestingly, we neither produced caproic acid in screening or long-term experiments. In addition to carboxylic acids, alcohols such as isopropanol and ethanol, which were both obtained in long-term experiments, are derived from acetyl-CoA [18].

#### 4.2.3. Electrofermentation by Thermococcales

As heterotrophs, Thermococcales do not fix CO_2_, but normally feed on organic carbon through glycolysis (Figure 6) [18]. In glycolysis, glucose is converted to pyruvate over a series of intermediates, including glycerol 3-phopshate and phosphoenolpyruvate, which is converted into pyruvate via a modified Embden–Meyerhof pathway [28]. However, neither screening nor long-term experiments were provided with a source of complex organics other than a low concentration of yeast extract (0.1 g/L) in the original medium. Instead, acetate was added as a nonfermentable potential source of carbon to Thermococcales. The metabolism of acetate is initialized by activation to acetyl-CoA as a precursor for catabolism (Figure 6) [18] by ACS enzymes (Appendix A), retrieved in Thermococcales [29]. None of the three Thermococcales strains we tested here have been grown on acetate before, but the continuous consumption throughout all experiments points to at least partial reliance on this substrate under electrotrophic conditions. Acetate consumption was complemented by formate metabolism in long-term experiments (Figure 6), as a variety of Thermococcales members can utilize one-carbon compounds such as CO and formate for growth via the reductive glycine pathway, using a pyruvate-formate lyase, although this yields very low energy [30,31,32]. Although the production of most compounds can be explained by autotrophy for the other strains, Thermoccocales cannot fix CO_2_ in a conventional way. Indeed, it was recently suggested that Thermococcales perform H_2_-dependent reduction of CO_2_ to formate to allow their survival in habitats depleted of complex organic compounds [33]. However, most of these compounds are also typical fermentation products (Figure 6) [18]. The Thermococcales strains may thus perform electrofermentation of acetate and potentially formate on the cathode. This way, acetate and formate are transformed into acetyl-CoA and pyruvate, producing small amount of ATP, later complemented with the conversion into organic acids and alcohols using the reducing equivalent produced from electron uptake from the cathode (potentially via hydrogenase) and producing more ATP (Figure 6). This metabolism was already suspected in community enrichment, where a strong correlation was observed between the growth of Thermococcales, high current consumption, and release of fermentation products from pyruvate [7]. Production of some of these products or the presence of enzymes involved in their production (Figure 6, Appendix A) has already been described in the literature for Thermococcales, such as for ethanol and isopropanol [34], propionate and lactate [35], alanine [29], succinate, or isobutyrate [36]. Alternatively, electron overfeeding from the cathode may lead to a release of intermediates that have never been observed in chemotrophic metabolism before.

#### 4.2.4. Respiration and Electron Acceptor Availability

Contrary to Thermococcales, autotrophic Archaeoglobales and Aquificales can only grow in the presence of a terminal electron acceptor that allows the synthesis of ATP for anabolic processes and the release of waste products from the cell in respiration (Figure 6) [37,38]. All strains used electron acceptors that were previously shown to support their growth, but clear preferences were established in most cases. Interestingly, these preferences sometimes varied from the inclinations observed in chemotrophic precultures. Contrary to GA, FP showed a strong response towards the soluble acceptors that were supplied as an alternative to iron, producing H_2_S via disproportionation in long-term experiments [39]. The excessive consumption of thiosulfate by this strain was not sustainable in the long term and contradicted the dependency on Fe (III) oxide under chemotrophic conditions. AF performed equally well on thiosulfate, but demonstrated a more moderate depletion, while both APy and AA appeared to rely largely on O_2_. In APy, this contradicted the preference for nitrate consumption in microaerophilic precultures (data not shown).

The coulombic efficiency for organic compound production varies significantly between the strains, reaching almost 100% for FP after CO_2_ was increased to 100%, and down to around 9% for MT at 100% CO_2_. For the respiratory strains (APy, FP, GA, and AF), the remaining electrons (not used for production of organics) are supposed to be transferred to an external electron acceptor via the quinone pool, leading to ATP production and sustaining the growth. However, this metabolism is subject to electron acceptor availability, which is mainly affected by the nature of the molecule (soluble or particular for iron oxide), the concentration, and the mass transfer to the cathode. Then, soluble electron acceptors, such as O_2_ for APy, and nitrate or thiosulfate for Archaeoglobales, are more susceptible to sustain growth on cathodes, leading to smaller coulombic efficiency for organic production than on a particular iron oxide. This is especially visible on FP between the phase at 20% CO_2_ with the supply of thiosulfate, leading to a CE for organic production around 52%, which increased to around 100% at 100% CO_2_ when the thiosulfate was replaced with iron oxide, diverting less reducing equivalent to respiration. Moreover, as ATP is required to fix CO_2_ in the autotrophic pathways, only survival is expected in absence of an abundant electron acceptor, explaining the low cell density in the biofilm at the end for FP. Indeed, only a small amount of ATP can be produced by the transformation of acetyl-CoA into organic products, using the reducing equivalent produced from the cathodic electron supply. Then, the lack of electron acceptor could force the cells to perform CO_2_ fixation into organic compounds, using CO_2_ as the electron acceptor to produce a small amount of ATP for survival. However, contrary to acetogens or methanogens, this metabolism might induce an energetic imbalance in the long term, which is not sustainable for the growth. Indeed, methanogens and acetogens have adapted their metabolism to be independent of external electron acceptors [40], producing small amounts of energy by methane or acetate production, leading here to low coulombic efficiency on side products and higher cell density over the long-term experiment for MT.

### 4.3. Biotechnological Application

The biotechnological application of these metabolisms requires high productivity, high coulombic efficiency, and, if possible, high product specificity. In long-term experiments, the combined CE_total_ reached up to 85% for GA, followed by FP, AF, TO, and down to 15–17% for MT and APy. However, as quantifications for gaseous (CH_4_, H_2_, and O_2_) compounds and ferric electron acceptors are missing, this only allows partial conclusions on the metabolic efficiency of these strains in MES. In addition, the partial evaporation of carboxylic acids at 75 °C and 55 °C may have influenced production rates, as observed in a control experiment (Appendix A). Then, the production of some organic acids and alcohols such as isopropanol and ethanol in our experiments were presumably underestimated, because of their volatility and low boiling point.

The production rates for each strain were averaged over 4 to 10 days of maximum production and reached to a maximum over the six strains: 3.97 g∙d^−1^∙m^−2^ of formate, 1.68 g∙d^−1^∙m^−2^ of acetate, 1.2 g∙d^−1^∙m^−2^ of succinate, 1.19 g∙d^−1^∙m^−2^ of propionate, 1.13 g∙d^−1^∙m^−2^ of glycerol, 0.89 g∙d^−1^∙m^−2^ of isopropanol, 0.75 g∙d^−1^∙m^−2^ of glycolic acid, 0.55 g∙d^−1^∙m^−2^ of butyrate, 0.50 g∙d^−1^∙m^−2^ of lactate, 0.41 g∙d^−1^∙m^−2^ of isobutyrate, and 0.23 g∙d^−1^∙m^−2^ of ethanol. Comparison of production rates is generally difficult due to non-standardized measurement units and differences in experimental systems (potential, material, or surface of the cathode), but approximations reveal that acetate, propionate, and isobutyrate production rates are similar to electrosynthesis performed previously, with 0.26 to 77 g∙d^−1^∙m^−2^ of acetate, 0.04 to 5.7 g∙d^−1^∙m^−2^ of butyrate, 0.05 to 0.18 g∙d^−1^∙m^−2^ of ethanol, and 0.06 g∙d^−1^∙m^−2^ of isopropanol, all obtained at more negative potential (from −0.8 to −1.1 V vs. SHE) [41]. Formate production in our system is interestingly 10 times higher than that previously reported of 0.36 g∙d^−1^∙m^−2^ at −0.8V vs. SHE [42]. This suggests that there is a “sweet spot” for maximal formate yields at elevated CO_2_.

Microbial electrosynthesis under hyperthermophilic conditions is a completely novel approach with global potential. We produced both succinic acid and glycerol, which are currently listed among the top 10 most valuable biobased products [43] as building block chemicals for the production of industrially relevant materials [44]. Succinic acid has a variety of applications, including the synthesis of specialized polyester and nylon precursors [43]. Thus far, succinic acid has mainly been derived from glucose fermentation by engineered strains of anaerobic bacteria [44]. Electrotrophic production in MES presents a significantly more energy-efficient route and may enable the substitution of expensive feed material by industrial CO_2_ emissions. In our experiments, the productivity rate for succinic acid (1.2 g∙d^−1^∙m^−2^) was highest in APy, followed by TO. Furthermore, glycerol is currently handled as the 10th most valuable biobased product, even though prices fell dramatically in recent years due to its side production in biodiesel fabrication. Nevertheless, it remains an important intermediate for the cosmetic, food, and pharmaceutical industries [22,43] and is generally promising in sustainable development due to its nontoxicity and biodegradability [44].

Finally, the effect of CO_2_ concentration on the metabolic shift observed in most strains is of great interest to potentially be able to tune organic production in specific pathways, as well as the optimization of purity and coulombic efficiency of the desired product. Other factors such as pH, temperature, or cathode potential are expected to have an additional impact, as demonstrated in alkaline conditions which can be expected to increase glycerol production. Furthermore, performing cocultures could enhance the production of certain compounds. For example, Thermococcocales may be paired with formate-producing Aquificales to facilitate electrofermentation of formate or acetate into propionic acid or alcohols.

## Figures and Tables

**Figure 1 microorganisms-10-02249-f001:**
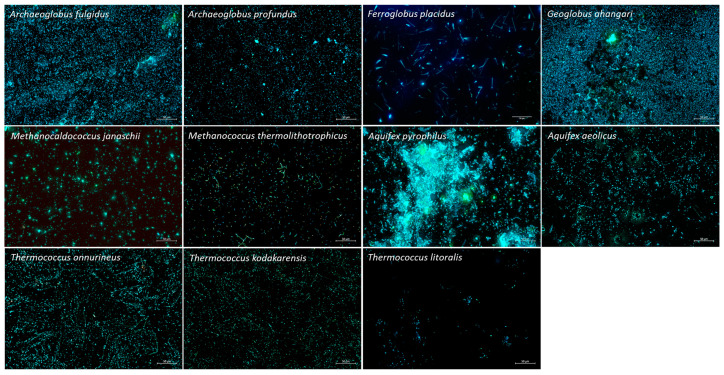
Biofilm formation on the graphite plate cathode during screening experiments as viewed under the fluorescence microscope with DAPI (**blue**) and Acridine orange (**green**). Scale bars are 50 μm.

**Figure 2 microorganisms-10-02249-f002:**
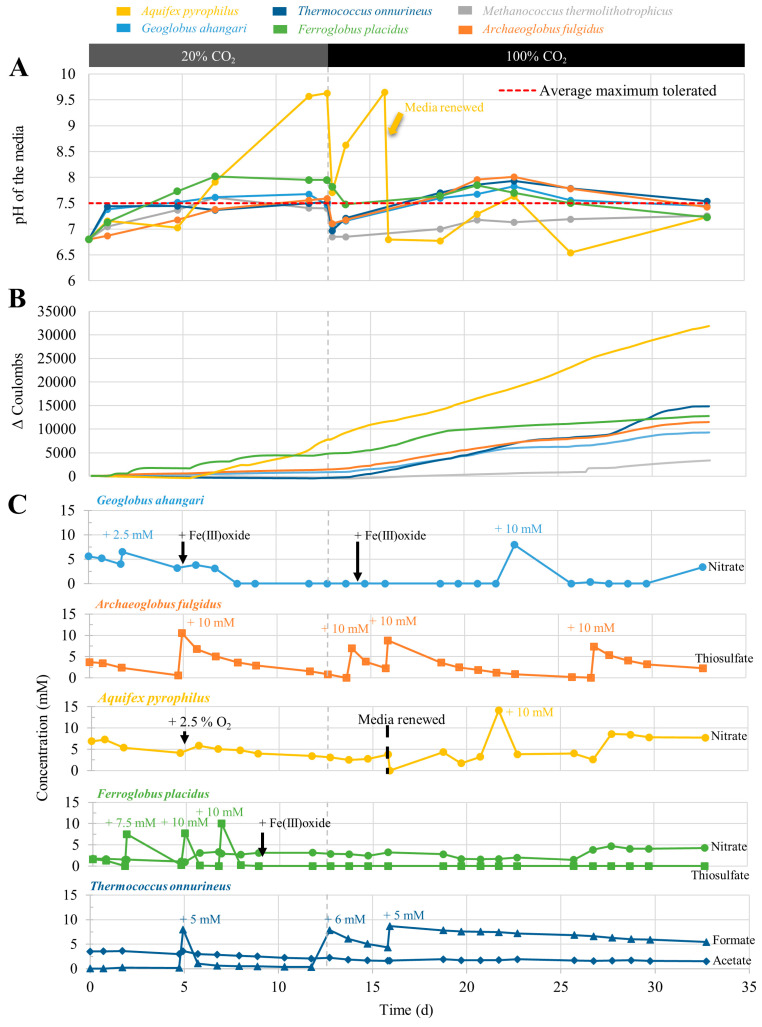
(**A**) The pH evolution; (**B**) current consumption between consecutive timepoints calculated by (I_biotic_ − I_abiotic_) ∗ (t_n_ − t_n−1_): for graphic depiction, Δ coulombs was subsequently averaged over 30 timepoints to normalize the curve; and (**C**) consumption of soluble electron acceptors in long-term experiments over time. CO_2_ was increased from 20% to 100% on day 13. In *Aquifex pyrophilus*, the media were renewed on day 16 due to high pH. Fe(III) reduction could not be quantified in this experiment due to partial sedimentation in the H-cell and O_2_ contamination of the samples.

**Figure 3 microorganisms-10-02249-f003:**
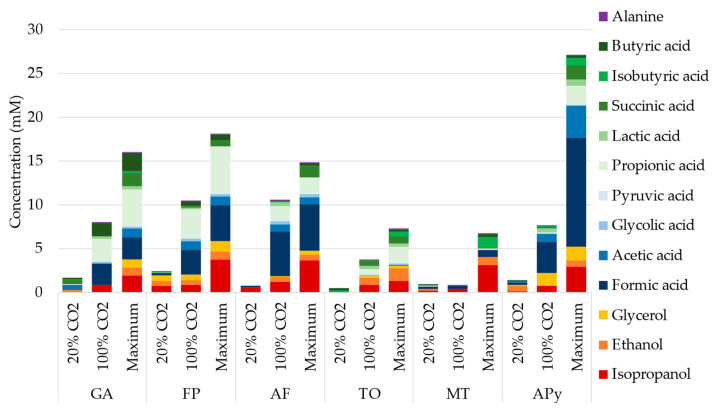
Organic molecule production measured by HPLC, NMR, and GC-MS during the 33 days of experiment with the 6 selected strains. The cumulative bar plots represent the production during the 13 days with 20% CO_2_ in the gas sparging, during the 20 days with 100% CO_2_ in the gas sparging, and the maximum concentration reached during all the experiments. The succinic acid and lactic acid were omitted for AF and FP at 20% due to overlapping peaks on HPLC data with the thiosulfate but added at 100% and maximum from NMR measurement on final sample.

**Figure 4 microorganisms-10-02249-f004:**
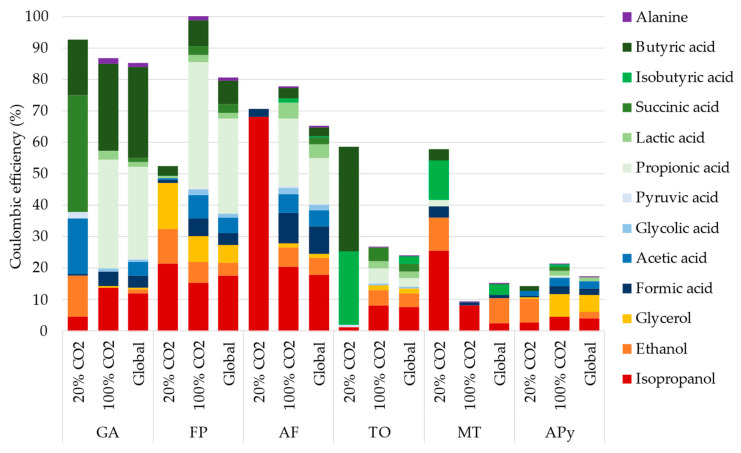
Coulombic efficiency of the organic production during the 33 days of experiment with the 6 selected strains. The cumulative bar plots represent the coulombic efficiency during the 13 days with 20% CO_2_ in the gas sparging, during the 20 days with 100% CO_2_ in the gas sparging, and the global coulombic efficiency during the whole experiment.

**Figure 5 microorganisms-10-02249-f005:**
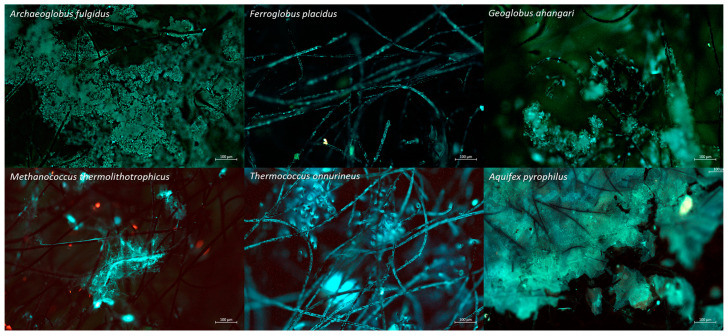
Biofilm formed on the graphite felt cathode during long-term experiments as viewed under the fluorescence microscope with DAPI (**blue**) and Acridine orange (**green**), both targeting DNA and RNA of the biofilm. The red color represents the autofluorescence of mineral/salt particles. Scale bars are 100 μm.

**Figure 6 microorganisms-10-02249-f006:**
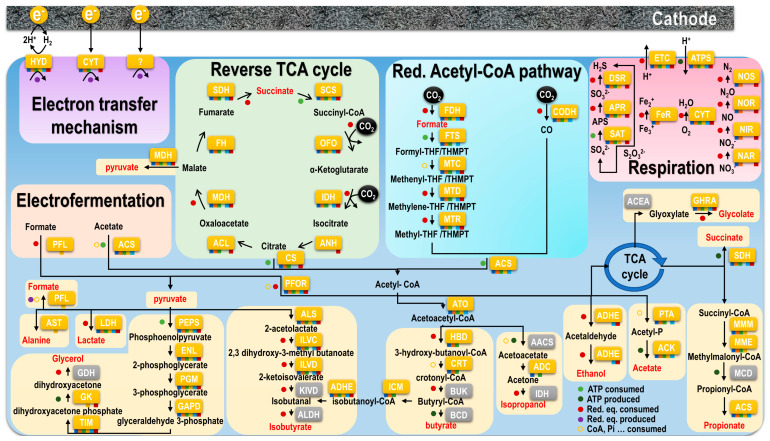
Hypothesized carbon metabolism at the cathode including the carbon fixation pathways of reductive acetyl-CoA pathway in Archaeoglobales and methanogens, reverse TCA cycle in Aquificales, and acetate and formate metabolism in Thermococcales, entering anabolic as well as catabolic processes including the use of different electron acceptors in respiration. Compounds yielded in our experiments are highlighted in red. Yellow labels represent the enzymes or equivalent enzymes involved in the pathways retrieved in the genome of the strains or close relative if not available and marked by a colored square if present for each strain (dark blue = GA, orange = FP, green = AF, yellow = TO, light blue = MT, and red = Apy). The grey labels represent enzymes not retrieved in the literature for those strains, suggesting an alternative pathway. For correspondence to complete name, please refer to Appendix A.

## Data Availability

Not applicable.

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
