# Peer review of "Screening for Hyperthermophilic Electrotrophs for the Microbial Electrosynthesis of Organic Compounds"

_microorganisms, 2022, doi:10.3390/microorganisms10112249_

Round 1

Reviewer 1 Report

The manuscript is based on a study of screening extremophiles for electrotrophic activity. The coupling of cathodic current as the source of electrons for microbial reductive metabolism is an upcoming field, and the manuscript highlights some prospects of this technology. The paper has minor errors and typos throughout the manuscript.

1.     33-35, 74. Two formats of literature citations.

2.     218, 278 didn’t – did not

3.     239. Syntax in ‘Inoculum were estimated’. Similar plural-singular errors also elsewhere in the text.

4.     Table 1. Specify the qPCR units in the caption or in the table itself.

5.     Table S1. Define X.

6.     Figure S1. 184-186. The graph has five strains instead of eleven strains. Define the letter codes in the caption.

7.     Figure S3.  mM.day-1 is not concentration.

8.     Chemical/biochemical compounds are not proper nouns and should not be capitalized; examples: Alanine, Nitrate; acetyl-CoA is written in three different versions in the manuscript.

9.     Figure 2C and elsewhere. It is not clear how the consumption of Fe(III) oxide as the electron acceptor was determined. What was the specific Fe(III) oxide used in this study?

10.  Figure 6: typos – consummed

11.  566. …the lack of electron acceptor could force the cells to perform CO2 fixation… Explain the underlying premise of this speculation.

12.  References. The format varies. 17. Nat. 2021 5927856 2021?  31. Nat. 2010 4677313 2010?  32. Microbiol. 2016 854 2016? 45. List this as Report DOE/GO-102004-1992. Scientific names should be italicized. Superscripts and subscripts should be noted. Single-word journal titles are not abbreviated. Check also #23, 34, 40, and 44.

13. Appendix A1 (line 303) was not in the review portal, but the information is presented in Figure S4 (supplementary Material).

Author Response

We thank the reviewer for their valuable comments and have improved the manuscript accordingly.

  1. 33-35, 74. Two formats of literature citations.

The citation has been modified to match the journal style

  1. 218, 278 didn’t – did not

The manuscript has been modified to correct this misspelling in the text.

  1. 239. Syntax in ‘Inoculum were estimated’. Similar plural-singular errors also elsewhere in the text.

The manuscript has been modified to correct this misspelling in the text.

  1. Table 1. Specify the qPCR units in the caption or in the table itself.

The qPCR unit has been added to the table.

  1. Table S1. Define X.

The X has been defined in the caption of the table.

  1. Figure S1. 184-186. The graph has five strains instead of eleven strains. Define the letter codes in the caption.

The legend formatting has been solved and the letter codes have been defined in the caption.

  1. Figure S3.  mM.day-1 is not concentration.

The axis label has been corrected to “Production rate”

  1. Chemical/biochemical compounds are not proper nouns and should not be capitalized; examples: Alanine, Nitrate; acetyl-CoA is written in three different versions in the manuscript.

The manuscript has been modified to correct these misspellings along the text.

  1. Figure 2C and elsewhere. It is not clear how the consumption of Fe(III) oxide as the electron acceptor was determined. What was the specific Fe(III) oxide used in this study?

The Fe(III) oxide has been defined in the Material and Method section (line 118) and a sentence was added in the Figure 2 caption to clarify that the Fe(III) reduction could not be quantified due to O2 contamination of the samples

  1. Figure 6: typos – consummed

The manuscript has been modified to correct this misspelling in the Figure

  1. 566. …the lack of electron acceptor could force the cells to perform CO2fixation… Explain the underlying premise of this speculation.

The lack of electron acceptor leads to a lack of ATP production to sustain the cell metabolism. To survive, the cell might use CO2 as electron acceptor, as for acetogens or methanogens, to produce small quantities of ATP. A sentence has been added to line 567 to clarify this point in the text.

  1. References. The format varies. 17. Nat. 2021 5927856 2021?  31. Nat. 2010 4677313 2010?  32. Microbiol. 2016 854 2016? 45. List this as Report DOE/GO-102004-1992. Scientific names should be italicized. Superscripts and subscripts should be noted. Single-word journal titles are not abbreviated. Check also #23, 34, 40, and 44.

The reference list has been revised as suggested

  1. Appendix A1 (line 303) was not in the review portal, but the information is presented in Figure S4 (supplementary Material).

Appendix A1 has been transferred in the caption of Figure S4 for clarity.

Reviewer 2 Report

In order to expand the number of microbial isolates for potential use in microbial synthesis and to provide knowledge of hyperthermophiles in this respect, Popall et al screened eleven strains previously isolated from hydrothermal vents. They performed two major set of experiments, one including all strains lasting for 6 days and one where they selected one representative for each and followed their growth on a for an extended period of time of 33 days. They observed production of biotechnologically relevant organic acid and alcohols under all test conditions and found that the strain grew on the catode as biofilms. Importantly the authors have expanded the collection of strains to be used in electrosynthesis in general and at high temperatures specifically. The work is presented and discussed in a very clear way and I have only a few minor comments:

1)    It is not easy to interpret the data presented in Figure 5. I would advise to add some arrows to guide the reader to where the biofilm is. 

2)    Check for acetyl-coA/acetyl-CoA

3)    Use the style of the journal for the reference list. There are also many typos.

Author Response

We thank the reviewer for their valuable comments and have improved the manuscript accordingly.

1)    It is not easy to interpret the data presented in Figure 5. I would advise to add some arrows to guide the reader to where the biofilm is. 

The caption of Figure 5 has been modified to clarify that all the blue/green signal corresponds to the biofilm.

2)    Check for acetyl-coA/acetyl-CoA

The manuscript has been modified to correct this misspelling in the text.

3)    Use the style of the journal for the reference list. There are also many typos.

The reference list has been revised as suggested